# *Antrodia Cinnamomea* Prolongs Survival in a Patient with Small Cell Lung Cancer

**DOI:** 10.3390/medicina55100640

**Published:** 2019-09-26

**Authors:** Huei Long, Chi-Tan Hu, Ching-Feng Weng

**Affiliations:** 1Department of Life Science and Institute of Biotechnology, National Dong Hwa University, Hualien 97401, Taiwan; longhuei@gmail.com; 2Research Centre for Hepatology, Hualien Tzu Chi Hospital, Buddhist Tzu Chi Medical Foundation, Hualien 97002, Taiwan; chitan.hu@msa.hinet.net; 3School of Medicine, Tzu Chi University, Hualien 97002, Taiwan; 4Department of Basic Medical Science, Center for Transitional Medicine, Xiamen Medical College, Xiamen 361023, China; 5Department of Biomedical Science and Environmental Biology, Kaohsiung Medical University, Kaohsiung 80708, Taiwan; 6Department of Food Science, National Kinmen University, Kinmen 89250, Taiwan

**Keywords:** *Antrodia cinnamomea*, small cell lung cancer, complementary medicine, prolong survival

## Abstract

*Introduction*: *Antrodia cinnamomea* (AC) is an extremely rare medicinal fungus native to forested regions of Taiwan. It possesses numerous biological activities, especially anti-tumor effects shown in various in vitro cancer cells and in vivo animal models. However, there are few clinical reports about AC as a treatment for cancer patients. This report attempts to demonstrate the therapeutic effect of dish-cultured AC (DAC) on a small cell lung cancer (SCLC) patient taken orally for an extended duration. *Patient concerns*: An 88-year-old male with a history of diabetes mellitus and hypertension visited the outpatient department with the symptoms of dyspnea and a cough for two weeks. After a diagnosis of SCLC, the patient declined both chemotherapy and radiotherapy because of the side effects and only accepted supportive care without additional therapy. *Diagnosis*: Limited-stage SCLC (T4N2M1a, stage IV) after the chest radiograph, computed tomography-guided biopsy, and pathological diagnosis. *Interventions*: The patient was prescribed DAC with an increasing dosage, from 5 g/d up to 10 g/d DAC, for six months, without radiation or chemotherapy treatment. *Outcomes*: DAC caused the tumor to shrink substantially. Surprisingly, the patient survived for 32 months without relapse after six months of DAC treatment. Laboratory examinations indicated that the patient’s health had improved significantly, reverting to near normal levels. Notably, he had a good quality of life with a high Barthel index score. Unfortunately, this patient died of septic shock caused by acute cholangitis. *Conclusion*: DAC may exert an anti-cancer effect, which can lead to tumor regression. This is supposed to be achieved by the combined DAC’s immunomodulatory, anti-angiogenic, anti-metastatic, anti-proliferative, and pro-apoptotic effects mediated through multiple signaling pathways. We propose that DAC can be used as a complementary medicine to prolong the life expectancy and improve the life quality of SCLC patients.

## 1. Introduction

Small cell lung cancer (SCLC) is the most aggressive of all lung cancer subtypes and comprises about 15% of lung cancer, while non-small cell lung cancer (NSCLC) comprises approximately 85% [1]. It has a dismal prognosis and is highly associated with smoking. Most patients can respond to primary therapy, but the survival rate is poor and the median survival time is reported to be around 14–20 months in cases of limited-stage disease and 7–10 months with extensive-stage disease, respectively [2]. The overall 5-year survival rates for SCLC in limited and extensive stages were reported to be only 12–17% and 2%, respectively [3]. Generally, SCLC results in mortality within 2–4 months without any treatment after diagnosis [4]. In limited-stage SCLC, concurrent chemoradiation therapy (CCRT) can be given to treat the tumor mass and lymph nodes. However, many patients with this late-stage disease choose to decline treatment because of the hardship caused by the side effects of CCRT.

In Australia, it is estimated that around 17% to 87% of cancer patients have used one form or another of complementary therapy during their cancer treatment [5]. There are numerous reasons and contributing factors for cancer patients to consider using complementary and alternative medicine (CAM), such as the physical and emotional distress associated with the diagnosis, the limited treatment options in the context of debilitating adverse effects from treatment, and the lack of substantial survival benefits in advanced cancers [5]. The frequent reasons for an average of 51% of cancer patients using CAM were improvement in their cancer and general health and a reduction in complications arising from the cancer or therapy [6].

Medicinal fungus *Antrodia cinnamomea* (AC) is a well-known Chinese folk medicine in Taiwan, known to possess numerous biological activities, especially an anti-tumor effect in in vitro cancer cells and in vivo animal models [7,8]. It is considered an efficient alternative phyto-therapeutic agent or an adjuvant to cancer treatment and immune-related diseases given its diverse bioactive compounds [9]. A total of 225 compounds have been isolated, identified, and structurally elucidated, including macromolecules (nucleic acids, proteins, and polysaccharides), small molecules (benzenoids, lignans, benzoquinones, and maleic/succinic acid derivatives), terpenoids (lanostane triterpenes, ergostane triterpenes, diterpenes, monoterpenes, and steroids), nucleotides (nucleobase and nucleoside), fatty acids, and fatty acid esters [10]. Cumulative in vitro and vivo studies have revealed its anti-diabetic and anti-hyperlipidemic [11], antihypertensive [12], anti-inflammatory [13], antioxidant [14], antimicrobial [15], cardiovascular disease preventive [16], immunomodulatory [17], hepatoprotective [18], and neuroprotective [19] effects. The anti-cancer effects of AC may involve the synergistic effects of a mixture of bioactive compounds in which each compound is responsible for its own specific molecular mechanisms, including the promotion of apoptosis, cell cycle arrest, anti-migration, anti-angiogenesis, and anti-inflammation [10]. Moreover, AC also acts indirectly against tumors via enhancement of the immune system by promoting a Th1-dominant state and natural killer cell activities through its unique polysaccharide component [20]. On the other hand, ethanol extracts of AC (EEAC) mycelia powder exerted anti-angiogenic effects by suppressing the phosphorylation of vascular endothelial growth factor receptor 2 (VEGFR2) and the expression of pro-angiogenic kinases in VEGF-treated human umbilical vein endothelial cells (HUVECs), in addition to reducing the expression of Janus kinase 2 (JAK2) and the phosphorylation of the signal transducer and activator of transcription 3 (STAT3) [21].

There are very few references to clinical research on the use of AC for cancer patients. Previously, a double-blind human trial (Trial registration: ClinicalTrials.gov NCT01,287,286) published in 2016 evaluated AC clinical effects in combination with chemotherapy [22]. In this study, eligible patients with stage III–IV adenocarcinomas (lung, breast, stomach, liver, or colorectal cancers), were scheduled to receive platinum-based chemotherapy and were subsequently randomly assigned to either the AC group or placebo group. The only significant difference found was in the patients’ platelet counts, which were lower in the AC group, and the sleep quality, which was also significantly improved [22]. However, the results did not show a significant difference in the mean 6-month survival rate. Nevertheless, this trial might have been premature because the treatment only lasted for 30 days. 

In this report, we demonstrate the anti-tumor effect, along with the health improving outcomes, of a patient administered only dish-cultured AC (DAC) for 6 months, which prolonged his survival time to 32 months compared with the average of 2–4 months in similar cases without DAC treatment [4].

## 2. Case Report

### 2.1. Case Report

The patient gave his informed consent for inclusion before participating in the study. The study was conducted in accordance with the Declaration of Helsinki, and approved by the Research Ethics Committee of Hualien Tzu Chi Hospital, Buddhist Tzu Chi Medical Foundation (Case report approval code: CR108-05, Form no. E6C0021106-03). The elected patient was an 88-year-old male with SCLC. All examination data were collected and analyzed, including chest radiograph, computed tomography (CT), and hematology tests. The patient’s wellbeing status was monitored throughout the study by regular interviews.

In December 2015, an 88-year-old male with a history of diabetes mellitus and hypertension visited the outpatient department at Jen-Ai Hospital–Dali Branch (Taichung, Taiwan) with the symptoms of dyspnea and coughing for 2 weeks. He had also lost 2 kg of body weight in the previous three weeks. He denied a fever or chills, chest pain, hemoptysis, nausea or vomiting, abdominal pain, diarrhea, or dysuria. Physical examination showed basilar crackles via chest auscultation. Laboratory data showed no leukocytosis (white blood cell, WBC count 7,520/uL). The segmented WBC was 72.6%. A chest radiograph revealed a tumor mass in the right lower lung field. A CT-guided biopsy was arranged for further pathological examination (Figure 1a). The pathological examination showed infiltrating carcinoma with a high nuclear-to-cytoplasmic ratio, nuclear molding, and increased mitotic activity (Appendix A). Immunohistochemistry revealed thyroid transcription factor 1 (TTF-1) (+), CD56 (+), chromogranin A (−), neuron-specific enolase (NSE) (−), and synaptophysin (weak +). In addition, the chest CT showed a huge tumor (17.26 × 9.78 cm in the largest dimension) in the right lower lung field, mediastinal lymphadenopathy, and right-sided pleural effusion (Figure 1a). The carcinoembryonic antigen (CEA) level was 18.4 ng/mL (reference values: male <3.4 ng/mL, smoking <6.2 ng/mL). The biopsy specimens were histologically diagnosed as SCLC in December 2015. As concluded by a Cancer Committee multi-disciplinary meeting, the clinical staging was stage IV (cT4N2M1a, limited stage). However, the patient declined both chemotherapy and radiotherapy due to associated potential side effects and only accepted supportive care without further oncologic therapies. The medical timeline is outlined in Figure 2

Subsequently, the patient agreed to take DAC (from 5 g/d up to 10 g/d) dissolved in water once daily for a total of 6 months, from February to July in 2016. The patient’s diet, medicine, and daily life were cared for by the patient’s wife during the period of DAC treatment. During this period, his symptoms of chest and back pain, shortness of breath, and a productive cough gradually improved. Therefore, he lived a decent quality of life, as indicated by the high Barthel index score (100/100). In July 2017, the follow-up chest CT scan showed that the tumor mass markedly reduced in size (8.89 × 3.90 cm in the largest dimension) (Figure 1b). After the period of DAC treatment, the patient did not display any symptoms of end-stage lung cancer, such as shortness of breath, chest pain, a cough, hemoptysis, or fatigue.

After taking DAC, his elevated fasting sugar and glycated hemoglobin levels returned to the normal ranges (Figure 3a), diastolic blood pressure (DBP) increased (Figure 3a), alanine transaminase (ALT) and aspartate aminotransferase (AST) levels were reduced, C-reactive protein (CRP) fell in the normal range (Figure 3b), serum creatinine decreased, estimated glomerular filtration rates (eGFR) resumed to the normal ranges (Figure 3c), segmented neutrophil counts dropped back to the baseline level, and low lymphocyte counts returned to the normal level (Figure 3d).

In July 2016, the patient had a major episode of depression upon his wife’s death due to a heart attack and unexpectedly ceased using DAC. Two years later (in August 2018), this patient died of septic shock caused by acute cholangitis, which was not associated with limited-stage SCLC. 

### 2.2. Methanol Extract Preparation 

Fruiting bodies of AC (synonyms: *Antrodia camphorata; Taiwanofungus camphoratus; Ganoderma comphoratus; Ganoderma camphoratum*) were collected from Yuli, Hualien County, Taiwan, in May 2012. The species was identified by Associate Professor Guan-Jhong Huang at the Department of Chinese Pharmaceutical Sciences and Chinese Medicine Resources, China Medical University, Taichung, Taiwan. A voucher specimen was deposited in the Department of Life Science and Institute of Biotechnology, National Dong Hwa University. AC was cultivated on artificial agar medium in a Petri dish. DAC raw powder was supplied by Longs Biotech Co., Ltd. (Hualien, Taiwan). The methanol (MeOH) extract of DAC raw powder was prepared following the unpublished modified procedure [23] by *Antrodia cinnamomea* Association of Taiwan Treasure (an independent Association in Taipei, Taiwan).

### 2.3. High Performance Liquid Chromatograph (HPLC) Profiles of Methanol Extracts of DAC

The metabolite profiling and quantification of a methanol extract of DAC was performed following the unpublished modified procedure [23] by *A. cinnamomea* Association of Taiwan Treasure. Briefly, the quantification of the content of each index compound in DAC was performed by HPLC analysis. The peak areas of the index compounds in the chromatogram of the methanol extract of DAC (with a known loading concentration, e.g., 100 ppm) were then defined, and amounts were calculated on the basis of the quantity calibrated from the internal standard (toluene or dexamethasone, 5 ppm) calibration curves.

HPLC profiles of the methanol extract of wood-state cultured AC (fruiting bodies) (Figure 4a), DAC (fruiting bodies) (Figure 4b), and DAC (fruiting bodies and mycelia) (Figure 4c) were evaluated and provided by *A. cinnamomea* Association of Taiwan Treasure. The methanol extract of DAC (fruiting bodies) showed a similar triterpenoids profile to wood-state cultured AC, which contained the same seven index compounds. DAC (fruiting bodies and mycelia) also has a similar triterpenoids profile to wood-state cultured AC, but a higher ratio of dehydrosulphurenic acid (DSA) and dehydroeburicoic acid (DEA) compared to the DAC (fruiting bodies). The seven index compounds characterized in wild AC fruiting bodies [23] and their detection limits are shown in Table 1.

## 3. Discussion

### 3.1. Patient Case

In general, stage IV SCLC results in mortality within 2–4 months after diagnosis without treatment due to its aggressive malignancy. In the case reported here, a patient with SCLC receiving only DAC for six months managed to survive for 32 months after clinical diagnosis. Furthermore, during the DAC treatment period, his chest and back pain, shortness of breath, and persistent cough gradually improved. Besides, an increase in body weight from 60.8 to 69.0 kg after the DAC treatment may suggest that the patient regained his normal physical condition (Figure 3a). 

The SCLC tumor of the patient was reduced in size after six months of treatment with DAC. DAC might inhibit the proliferation of SCLC cells and promote cell apoptosis due to antrocin, an active compound from AC fruiting bodies, which increases the level of cleaved caspase-3 and the Bax/Bcl2 ratio and downregulates the JAK/STAT signaling pathway through increasing microRNA let-7c expression [7]. Additionally, the sulfated glucan from AC mycelia (SGA) might also suppress SCLC tumor cell viability and migration through an inhibitory effect on the TGF-β/FAK/AKT axis [24]. Most patients with SCLC relapse within a year upon treatment [3]. In this case, the patient survived without relapse for over two years after six months of treatment with DAC. If any tumor activity remains after treatment with DAC, a patient with SCLC will relapse within a year due to the rapid doubling and widespread metastases of SCLC. Moreover, SGA also potentiated cisplatin-induced cytotoxicity in lung cancer cells [24]. This implies that DAC may also act as an adjuvant for SCLC patients undergoing cisplatin chemotherapy.

The patient with a history of diabetes mellitus (type II) resumed normal blood sugar and glycated hemoglobin levels from 140 to 94 mg/dL and from 7.2% to 6.2%, respectively, from first being diagnosed of SCLC until the end of DAC treatment. In comparison with the prior two years without DAC treatment, the patient’s fasting blood glucose and glycated hemoglobin level ranged between 127 and 107 mg/dL and between 7.6% and 6.6%, respectively (Appendix A). Given that the patient’s dietary intake did not change during that period, the improvement can most likely be attributed to AC’s anti-diabetic effect due to its several anti-diabetic compounds, such as DEA, antcin K, eburicoic acid, DSA, and antcin B [11,25,26,27]. 

Regarding the improvement of the liver-renal function in SCLC patients, AC mycelia have been reported to lower oxidative stress associated with reducing the expression of cleaved caspase −3, −8, and −9, and the levels of phosphor-protein kinase B (Akt) and phosphor-nuclear factor-κB (NF-κB) in the liver [28]. Antcin B and antcin K, the compounds from AC fruiting bodies, can exert a dose-dependent decreasing effect on the serum levels of ALT and AST, and decrease the incidence of liver necrosis through an anti-inflammation mechanism associated with down-regulating IL-1β, TNF-α, iNOS, COX-2, and NF-κB in liver tissues [29]. Antcin H from AC fruiting bodies can protect against liver injury through disruption of the binding of p-JNK to Sab, an outer membrane p-JNK receptor and substrate, which interferes with the reactive oxygen species (ROS)-dependent self-sustaining activation of the mitogen-activated protein kinases (MAPK) cascade [30]. Antroquinonol, a component from AC mycelia, has been found to ameliorate hypertension and improve the renal function by reducing oxidative stress and inflammation. It decreased pro-inflammatory cytokine concentrations in the kidney by modulating the NF-κB pathway [31].

During the period of DAC treatment, the diastolic (DBP) and systolic blood pressure (SBP) elevated from 52 to 64 mmHg and from 102 to 106 mmHg, respectively. Moreover, after the period of DAC treatment, the DBP returned from 64 to 59 mmHg. However, his SBP elevated from 106 to 112 mmHg (Figure 3a). The differences between systolic and diastolic pressure during the period of DAC treatment were decreased in contrast to the differences between systolic and diastolic pressure during the period without DAC treatment. This may suggest an increase in the arterial wall elasticity during the period of DAC treatment, although increased DBP may be a potential side effect of DAC.

In 2016, a clinical study evaluating therapeutic effects of AC in advanced cancer patients receiving standard chemotherapy concluded that platelet counts were significantly lower in the AC group than in the placebo group after 30 days of treatment. However, we did not observe any evidence that DAC interferes with the processes of platelet production as the patient’s platelet counts had not shown obvious change after six months of DAC treatment (Appendix A). 

### 3.2. What Factors Influence the Efficacy of Using AC?

In 2016, a clinical study revealed that AC did not affect the progression of tumor sizes in gastric, lung, liver, breast, and colorectal cancer when combined with chemotherapy after a short treatment period that only lasted for 30 days [22]. They postulated that the necessity of a long-term treatment may show the efficacy of AC. In this study, our patient received AC treatment for a longer period, without the side effects of chemotherapy. Instead, we demonstrated the anti-tumor effect after a long period using DAC (six months). Notably, the patient with a history of diabetes mellitus (type II) and hypertension resumed a normal blood sugar level, renal functions, and neutrophil and lymphocyte counts, as well as liver functions within the normal range. The aqueous extract of AC mycelia used in the human trial was mainly comprised of 2100 mg polysaccharides, 172 mg triterpenoids, and 2687.5 mg γ-aminobutyric acid. In addition, the participants in the clinical trial of a much shorter duration were given an aqueous extract in 20 mL oral formulations twice a day for 30 days. By contrast, our patient was only given 10 g of DAC once a day for six months. The raw powder of DAC consists of polysaccharides, benzenoids, and index compounds that account for several biological activities, such as immunomodulation, anti-cancer activity, anti-inflammation activity, anti-diabetes activity, hepato-protection, and renal-protection. However, after ceasing the use of DAC, the maintained conditions of the patient’s serum biochemistry gradually relapsed, after three months to one year (Appendix A). Despite this, DAC apparently exerted an anti-tumor effect on the SCLC patient during long-term therapy. A long-term intake of DAC appears beneficial, particularly in patients with SCLC. We hypothesize that the tumor-inhibiting effect of DAC, along with the health benefits supported by hematological and biochemistry readings, contribute to the improvement of progression-free survival. Additionally, the safety of DAC is also attested by the patient’s improved liver and renal functions. 

### 3.3. HPLC Profiles of a Methanol Extract of DAC 

In Taiwan, AC is an extremely rare naturally occurring fungus and considered the most expensive herbal medicine because of the high demand for it. Currently, AC can be cultivated by multiple methods, including wood-state cultivation, Petri dish cultivation, solid-state fermentation, and submerged fermentation. Culture conditions, such as cultivation methods, temperature, pH, and time, can significantly affect the composition and content of bioactive components, which, inadvertently affect the therapeutic impact of the cultured product. Wood-state cultivation is the method closest to the growing condition of wild AC mushrooms. Triterpenoids are abundant and considered the major active components of AC fruiting bodies responsible for numerous pharmacological and therapeutic effects [32]. Our results showed that DAC contains an index compound profile very similar to that of wood-state cultivated AC fruiting bodies. Our results showed that DAC could be considered a reliable mode of propagation for meeting the increasing demand for wild AC mushrooms.

## 4. Conclusions

In this SCLC patient, the unique treatment significantly reduced the tumor size by using DAC up to 10 g/day for six months. The effects of DAC seem to prolong survival without relapse and to improve the quality of life through its anti-diabetic, hepatoprotective, and reno-protective functions. Although, in this case, we could not demonstrate its immunomodulatory, anti-angiogenic, anti-metastatic, anti-proliferative, and pro-apoptotic effects, as in the literature, the patient’s dramatic improvements might be attributable to these mechanisms. Above all, we have shown that DAC may be used as a complementary medicine with multiple mechanisms to reduce tumor size and improve progression-free survival in a patient with SCLC. One case, of course, is not enough to establish a definite conclusion. More clinical studies are needed to confirm DAC’s effects on lung cancer. To the best of our knowledge, this is the first case report showing the therapeutic effect of DAC in an SCLC patient.

## Figures and Tables

**Figure 1 medicina-55-00640-f001:**
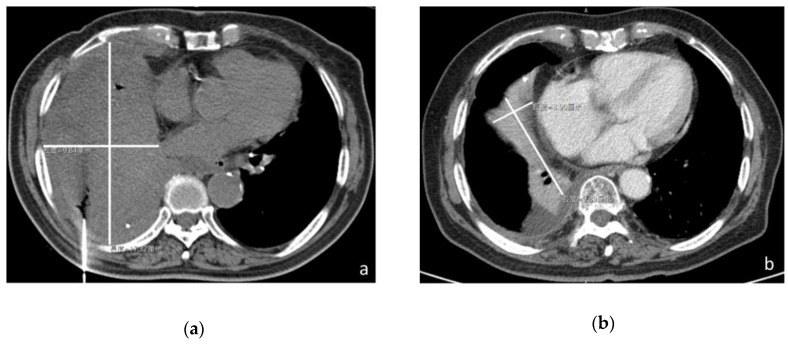
Dish-cultured *Antrodia cinnamomea* (DAC) shrinks the tumor in a cancer patient with administration for only 6 months. CT image of the patient (**a**) in December 2015 revealed a huge tumor (17.26 × 9.78 cm in the largest dimension) in the right lower lung and (**b**) in July 2017 showed the tumor mass remarkably reduced in size (8.89 × 3.90 cm in the largest dimension).

**Figure 2 medicina-55-00640-f002:**
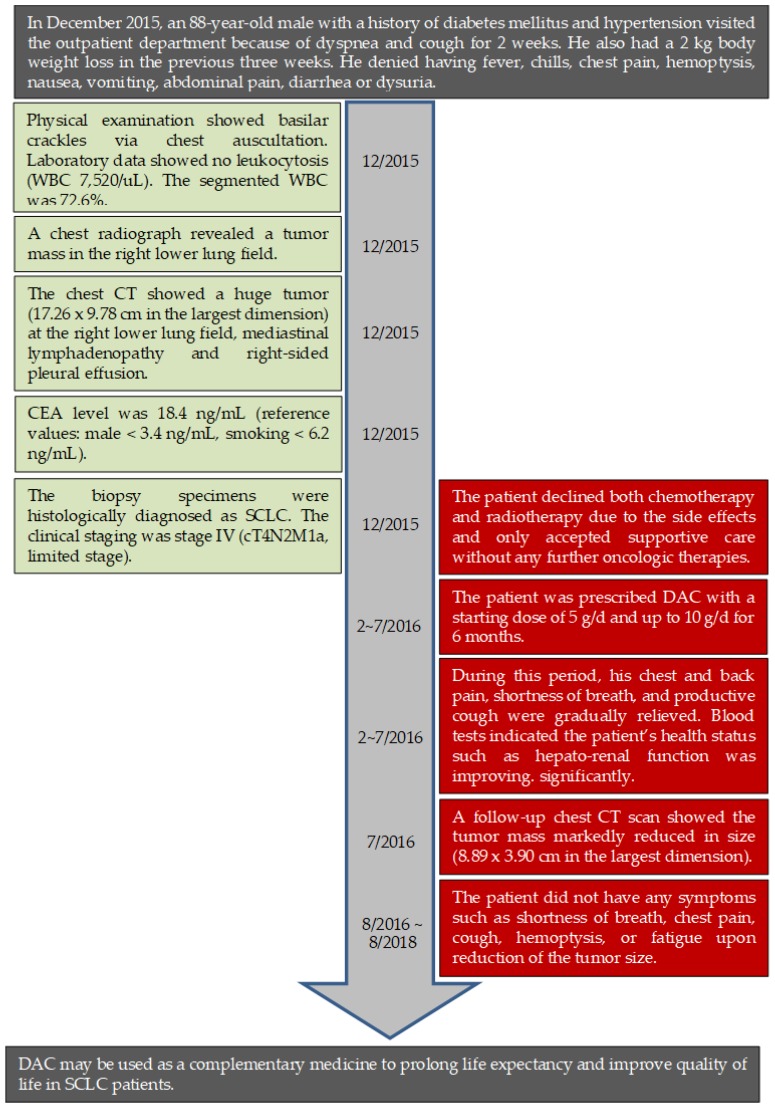
Timeline of interventions and outcomes. The left green boxes describe the clinical presentations and clinical diagnosis while the red boxes in the right side reveal the interventions and outcomes.

**Figure 3 medicina-55-00640-f003:**
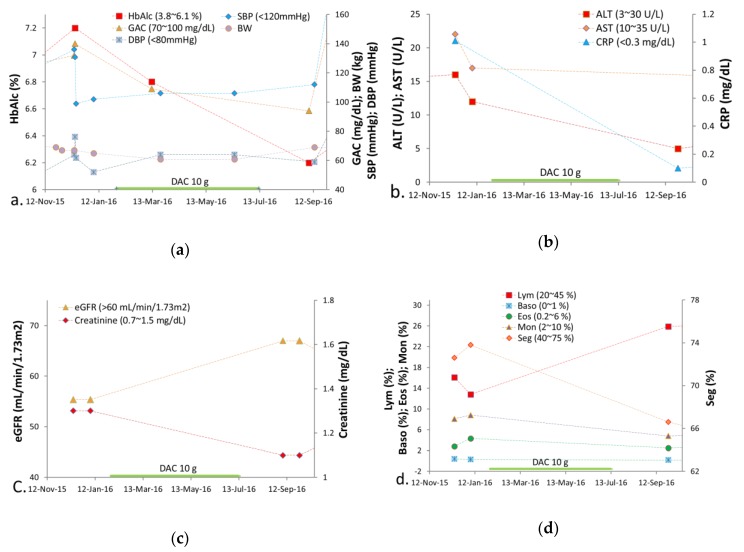
The findings of biochemistry examinations and hematology test of the small cell lung cancer (SCLC) patient during the DAC treatment period. (**a**) The elevated HbA1c returned to the normal range. The high fasting sugar levels also declined to the normal range. (**b**) The ALT and AST levels were improved. The CRP levels returned to the normal range. (**c**) The serum creatinine levels decreased. Low eGFR increased to the normal range. (**d**) The Seg returned to the baseline level. Low Lym counts regained the normal level. The Mono, Eos, and Baso counts remained stationary. GAC = glucose ante cibum (before meals); HbA1c = glycated hemoglobin; BW = body weight; SBP = systolic blood pressure; DBP = diastolic blood pressure; ALT = alanine transaminase; AST = aspartate aminotransferase; CRP = C-reactive protein; eGFR = estimated glomerular filtration rates; Seg = segmented neutrophil counts; Lym = lymphocyte; Mon = monocyte; Eos = eosinophil; Baso = basophil.

**Figure 4 medicina-55-00640-f004:**
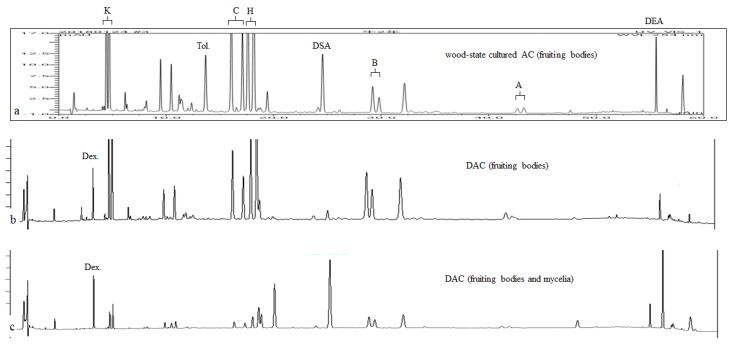
High Performance Liquid Chromatograph (HPLC) profiles of a methanol extract of cultivated *Antrodia cinnamomea* (AC). (**a**) Wood-state cultured AC (fruiting bodies); (**b**) DAC (fruiting bodies); (**c**) DAC (fruiting bodies and mycelia). K = (R, S)-antcin K; C = (R, S)-antcin C; H = (R, S)-antcin H; B = (R, S)-antcin B; A = (R, S)-antcin A; DSA = dehydrosulphurenic acid; DEA = dehydroeburicoic acid; AC = *Antrodia cinnamomea*; Tol. = toluene (internal standard); Dex. = dexamethasone (internal standard).

**Table 1 medicina-55-00640-t001:** The seven index compounds characterized in wild AC fruiting bodies and their detection limits.

	Index Compounds	Detection Limit (mg/g)
Ergostane-type triterpenoids	(R,S)-antcin K	0.002
	(R,S)-antcin C	0.002
	(R,S)-antcin H	0.004
	(R,S)-antcin B	0.002
	(R,S)-antcin A	0.007
Lanostane-type triterpenoids	DSA	0.003
	DEA	0.002

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
