# Peer review of "Antrodia Cinnamomea Prolongs Survival in a Patient with Small Cell Lung Cancer"

_medicina, 2019, doi:10.3390/medicina55100640_

Round 1

Reviewer 1 Report

The authors of the manuscript “Antrodia cinnamomea prolongs survival in a patient with small cell lung cancer" reported a case of an 88-year-old male with limited-stage small cell lung cancer (SCLC) using Antrodia cinnamomea (AC) only for six months. The study claimed that AC may have diverse therapeutic effects, such as anti-tumor activity, anti-diabetic activity, hepatoprotective effect and reno-protective effect, to improve the symptoms of SCLC and quality of life, without any side effects.

Main concerns:

1. In terms of the materials, a voucher specimen number should be provided in a botanical study, as vouchers are critical for authenticating and identifying a taxon (type specimen). It was used to confirm identification and verifications for reliability and reproducibility of scientific investigations (e.g. Mu Tong (caulis akebiae; celmatis armandii; Aristolochia manshurensis)). A qualified person should be recruited to identify the sample raw materials, since misidentification of sample may lead to unreliable of study results and therefore, hamper drug development.

2. Limitations of this study should be discussed. The effects of AC for SCLC were only confirmed in this single case report and the other clinical trial. Thus, the evidence of the efficacy of AC is not concrete enough that may mislead practitioners in clinical practice.

3. In section 1, paragraph three line one, Abbreviations of AC should be defined in parentheses the first time they appear in the main text.

4. In section 2.3, the index compounds should be list clearly, as well as the minimal level of each compound in AC, to confirm the materials were the correct taxon to be used.

5. In section 4.1, paragraph one line five, the authors indicated they increased the dose of AC to 10 g/d, however, they never mentioned the initial dose before. I am wondering whether this is a typo error.

6. In section 4.2 paragraph one the sixth line from the bottom, the authors believed that dish-cultured AC (DAC) is necessary for patients with cancers. Nonetheless, the whole manuscript only mentioned DAC may be beneficial for lung cancer patients. It seems to exaggerate the treatment effects of DAC.

7. Figure 2 and Figure S1 are difficult to read, a line chart could be used.

Reviewer 2 Report

Authors present a case report about a long surviving SCLC patient treated with the natural extract Antrodia cinnamomea only. Data are presented in a confused way, it is not clear if it is a case report only or a trial which implies ethical concerns. The figures of blood analyses are unclear and the conclusions are too much strong. They must be mitigated. 

Reviewer 3 Report

The case report by Long et al and colleagues provide interesting observations in regards to the therapeutic potential of Antrodia cinnamomea extracts in the treatment of late-stage cancers. The compound resulted in remarkable tumor shrinkage and improvement in several biochemical markers during 6 months treatment period. The report is nicely constructed, however, certain points need to be addressed because it is finally accepted for publication.

Major comments

-The manuscript requires major language revision, preferably, by a native English speaker before it is appropriate for submission.

-I suggest including histopathological data indicating that the observed thoracic mass is SCLC as that would enrich the report.

-Figure 2 (and similar supplemental figures) nicely reflect the changes in the biochemical profile of certain markers, however it is difficult to read, and the authors are encouraged to make the labels larger. Example: it is not clear if AST was actually reduced from looking at figure 2b.

-The authors highlight the decrease in HbA1C levels, which is not very significant ~ 1% within 6 months without any comments on how the patient’s dietary intake has changed during that period. The authors should be more careful in attributing their findings to the sole effect of AC. Moreover, using HbA1C as a lone indicator for the anti-diabetic effect is not sufficient and the authors are encouraged to use a more conservative language.

-The authors are encouraged to mention the unfavorable effects of AC in the Results section e.g., the obvious elevation in DBP. Increased blood pressure can be a potential side effect of AC.

-In the Conclusions section, the authors mention that AC had anti-metastatic, anti-proliferative and pro-apoptotic effects. The authors need to show evidence for this statement; otherwise, it is not supported by the presented data.

-The authors explain why their results are different from previous clinical trials that failed to show an antitumor effect of AC in combination with chemotherapy. The authors should provide their insights on future studies and how a more valid clinical trial on AC should be conducted.

Minor comments

-Several sentences in the text need proper rewording. Example: The sentence “An 88-year-old male with limited-stage SCLC (LS-SCLC, T4N2M1a, stage IV) after the pathological 21 diagnosis took 10 g daily administration for 6 months without any chemotherapy treatment, AC 22 caused the tumor shrinks of SCLC patient.” in the Abstract.

- LS-SCLC needs to be defined in the text.

Reviewer 4 Report

This is a very interesting and encouraging case report, which deserves publication. I congratulate to the result. However, you cannot draw conclusions from a single case; there might have been a spontaneous healing anyway. Therefore, throughout the manuscript, rewrite about DAC, that the healing occurred during DAC treatment and it appears that DAC has led to the positive result. E.g. line line 27 “It may be suggested that AC exerted …. “ or line 308: “It seems (appears)  to prolong ….”

I encourage performing a double blind study to strengthen the results.

Line 90: Please rewrite: “The patient gave his informed consent for inclusion before he participated in the study.”

Line 94: Please omit: “The patient who provided a 94 written informed consent form to participate in this study.”

Lines 95, 96: Please provide town and country of “Longs Biotech Co., Ltd.”

Lines 98, 99: “Patient’s wellbeing status was 98 monitored through structured interview.”: Why did you not use validated questionnaires such as EORTC-QLQ-C30 or SF-36? Please explain.

Line 103: “Antrodia cinnamomea Association of Taiwan Treasure”: Is this an independent Association or a private institution? Please provide more details, also address (in a footnote or reference) if possible.

Line 108: “HPLC”: Provide full name before abbreviation in parenthesis.

Line 115: Please provide town and country of “Jen-Ai Hospital–Dali Branch”.

Line 123: Please provide reference values for CEA.

Line 132: Did he take the powder dissolved in water? Please clarify.

Line 145: Why is CEA not shown?

Lines 159ff: I think para 3.2 should move to para 2.2 in the Methods section as it is no result.

Lines 180 to 182: Suggestion to rewrite: “Furthermore, during the treatment period with only DAC, after the patient was prescribed DAC with increasing doses up to 10g/d, his chest and back pain, shortness of breath, and productive cough were gradually improved.”: ok?

Line 274: Progression of which tumor entity?

Round 2

Reviewer 2 Report

Authors have partly addressed the comments. I suggest to reduce the description of the effects of AC (paragraphs 3.2 to 3.5) because they are basically a review of the literature which is out the scope of the paper.

English should be revised.
